# The inequality labor loss risk from future urban warming and adaptation strategies

Cheng He [1,2,3], Yuqiang Zhang [4], Alexandra Schneider [5], Renjie Chen[3], Yan Zhang[1,6,7,8], Weichun Ma[1,6,7,8,9 ✉], Patrick L. Kinney[2 ✉] & Haidong Kan [3 ✉]

Heat-induced labor loss is a major economic cost related to climate change. Here, we use hourly heat stress data modeled with a regional climate model to investigate the heat-induced labor loss in 231 Chinese cities. Results indicate that future urban heat stress is projected to cause an increase in labor losses exceeding 0.20% of the total account gross domestic product (GDP) per year by the 2050s relative to the 2010s. In this process, certain lower-paid sectors could be disproportionately impacted. The implementation of various urban adaptation strategies could offset 10% of the additional economic loss per year and help reduce the inequality-related impact on lower-paid sectors. So future urban warming can not only damage cities as a whole but can also contribute to income inequality. The implication of adaptation strategies should be considered in regard to not only cooling requirements but also environmental justice.

[1] Department of Environment Science and Engineering, Fudan University, Shanghai, China. [2] School of Public Health, Boston University, Boston, MA, USA. [3] School of Public Health, Key Lab of Public Health Safety of the Ministry of Education and NHC Key Lab of Health Technology Assessment, Fudan University, Shanghai 200032, China. [4] Gillings School of Global Public Health, University of North Carolina at Chapel Hill, Durham, NC, USA. [5] Institute of Epidemiology, Helmholtz Zentrum München – German Research Center for Environmental Health (GmbH), Neuherberg, Germany. [6] Institute of Digitalized Sustainable Transformation, Institute for Big Data (IBD), Fudan University, Shanghai, China. [7] Shanghai Key Laboratory of Atmospheric Particle Pollution and Prevention (LAP3), Fudan University, Shanghai, China. [8] Shanghai Institute of Eco-Chongming (SIEC), Shanghai, China. [9] Shanghai Key Laboratory of Policy Simulation and Assessment for Ecology and Environment Governance, Shanghai, China. ✉email: wcma@fudan.edu.cn; pkinney@bu.edu; kanh@fudan.edu.cn

Future increases in temperature over the next few decades in response to increasing concentrations of greenhouse gases have been projected at the regional[1,2] and global[3] scales. Increased temperatures have been demonstrated to generate increasingly severe limitations on human activity and health, especially in urban areas[4,5]. Moreover, global population increases are projected in urban areas, which will increase urban heat risk[2] and expose more people to projected warming[6]. Adaptation to urban warming is becoming increasingly important, as cities are and will continue to be the primary residential areas and workplaces; this is especially true in China, where more than 19.74% of the world's urban population currently lives, and more than 70% of the projected population increase will occur in urban areas[7]. Therefore, choosing cities in China as examples could reveal the considerable impact of continued urbanization on urban warming in the future.

Another important but overlooked aspect of climate change is the global-scale increase in absolute humidity[8]. The extreme apparent temperature, a heat stress index incorporating humidity and temperature, rises much faster than does the air temperature over land[9]. Under extremely high temperatures and high humidity levels, heat stress can occur as a result of an insufficient ability of the human body to dissipate heat, leading to heat exhaustion, heat stroke[10], and aggregate effects that could reduce labor productivity[8,9]. Several studies have indicated that increased heat stress in the workplace due to climate change can significantly impact occupational safety[11–14]. Moreover, studies have determined that the economic cost arising from the above labor loss is higher than any other related impact of climate change[15,16].

Most studies aimed at estimating the labor loss due to future warming adopt a large-scale perspective, focused on the global[17] or country scale[18]. While providing critical estimates of the relationship between climate change and labor loss, these studies fall short in evaluating the degree to which actual labor is affected at the regional scale, especially in different cities. Due to their varied total populations, age patterns, and economic structures, the adaptive capacities of cities to climate change differ and have not been fully considered in current studies. Under continuous urbanization, rural residents are increasingly migrating to cities, and the urban economy may be increasingly affected. However, these effects and the consequential environmental damage to individual cities have rarely been addressed because the local climate conditions at the city level are more complex and influenced by more factors than the climate conditions on a large scale. Furthermore, the urban economy is highly complex and involves numerous sectors. The aim of our paper was to address these gaps.

Moreover, as urban heat stress is more relevant to sectors involving more outdoor work and lower pay, such as construction and manufacturing, than to other sectors, the distribution of the economic damage due to future urban warming raises environmental justice concerns. This issue requires special attention, as many new immigrants from the countryside to urban areas are likely to be employed in these labor-intensive but climate-sensitive sectors. Furthermore, new migrants to cities, especially cities in North China, usually live in affordable housing units without air conditioning (AC) systems, leading to further adverse impacts. Our study considered these complicated aspects of urban warming to provide necessary guidance for urban design aimed at environmental justice in the foreseeable future.

In this work, with the use of a regional climate model coupled with an urban canopy model (UCM), we aimed to dynamically downscale climate change scenarios involving shared socioeconomic pathways (SSPs) to the local scale to quantify the potential influences of the interactions among climate change and potential urban adaptation strategies (the adoption of green roofs, cool walls, and cool ground surfaces) on future urban warming, measured as the hourly wet-bulb global temperature in shadow (WBGTs), a heat stress metric incorporating temperature, humidity, and other environmental factors; this metric exhibits broad occupational health applications and has been suitably validated against industrial[19,20] and US military labor standards. Next, based on the widely used exposure–response function (ERF) method[11,21], we assessed the economic costs attributed to urban heat. To address the influences of the varied work intensities in different sectors, we divided the sectors in each city by the level of work intensity in economic cost assessment. This study covered 18 types of sectors in 231 cities across mainland China. Eight urban agglomerations with different socioeconomic and climatic backgrounds were selected as key analysis units. The results of this study could provide valuable insights into the evaluation of the cost of climate change at the urban scale and could inform the selection of urban climate adaptation strategies.

## Results

**Future urban warming patterns with and without adaptation.** Our modeling results suggest that without the implementation of any adaptation strategies, most of the cities across mainland China will undergo significant heat increases by the 2050s (Fig. 1). The average WBGTs value during the possible working times in summer ($T_{area}$) across all urban grids is projected to increase by 1.80 ± 0.32 °C and 2.05 ± 0.33 °C under representative concentration pathway (RCP) 6.0 and RCP 8.5, respectively. The change in the regional population-weighted WBGTs ($\Delta T_{pop}$) is more than 0.12 °C greater than the change in $T_{area}$ ($\Delta T_{area}$) under these two scenarios (Fig. 2), indicating that the influence of the population distribution on heat stress will extend further with urbanization. Regionally, $\Delta T_{area}$ is projected to increase along the coast from northern Shandong to southern Zhejiang, as the high humidity in this region results in surface heat conditions that are more sensitive to the enhanced radiative forcing under future climate change. In contrast, significant WBGTs growth under RCP 6.0 is projected in the eastern coastal region, while severe warming under RCP 8.5 is projected not only along the eastern coast but also across northeastern China. This spatial pattern is consistent with the results of recent studies performed at the global scale[3] and within mainland China[22].

Motivated by recent carbon-mitigation efforts worldwide, many countries have released specific timelines and paths to achieve carbon neutrality. Therefore, we believe that carbon emissions will be controlled worldwide in the foreseeable future. Against this background, we selected RCP 6.0 as the scenario to investigate the implementation of adaptation strategies. As shown in Fig. 2 and Supplementary Fig. 2, the modeling results suggest that the abovementioned three adaptation strategies can cool the urban environment, with significant regional variability. In most regions, these cooling strategies are projected to reduce the average $T_{area}$ value by more than 0.20 °C. However, higher average warming of 1–2 °C persists across the remainder of mainland China, broadly consistent with recent research in other country[23,24]. Cooling effects are the strongest along the coast from Shandong in the north to Guangdong in the south, as most larger cities are concentrated in this area. Cooling effects are weaker in the Chengyu region (CY) and the middle reaches of the Yangtze River (MYR). Because these regions exhibit a higher cloud coverage during summer than that in other regions, which means less solar radiation can be influenced by these adaptation strategies. Heat pressure is projected to increase in small cities in northeastern China following the adoption of these urban strategies. As these urban adaptation strategies could weaken urban circulation, moist air that would otherwise rise could

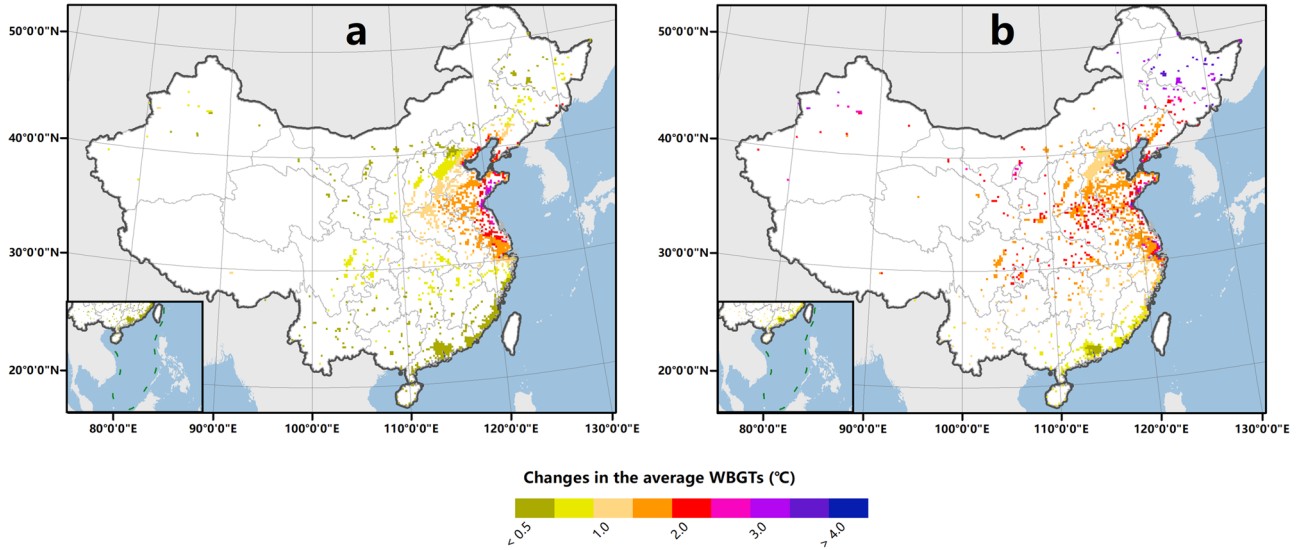

**Fig. 1 Changes in the average wet-bulb global temperature in shadow (WBGTs) value during the possible working times (8:00–20:00) in summer by the 2050s relative to the 2010s.** The associated scenarios are RCP 6.0 (**a**) and RCP 8.5 (**b**). Each point indicates the value in each urban grid (the urban land-use fraction is projected to be higher than that of any other land-use type in each grid by the 2050s) at a 20-km resolution.

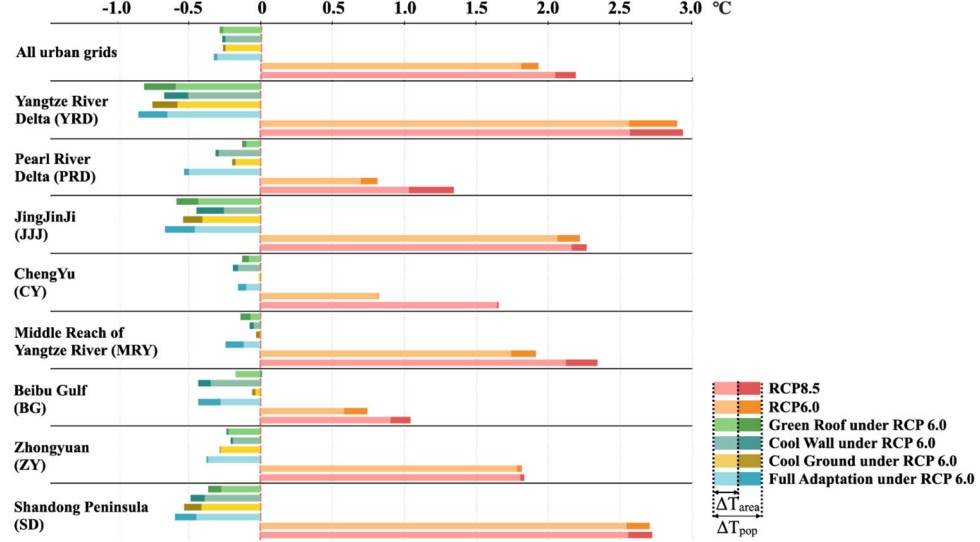

**Fig. 2 $\Delta T_{area}$ and $\Delta T_{pop}$ resulting from future changes and individual adaptation strategies.** $\Delta T_{area}$ denotes the average change without considering the distribution of urban residents. $\Delta T_{pop}$ denotes the risk change incorporating the distribution of residents, which is obtained by weighting the local average heat level by the corresponding number of urban residents within all urban grids.

remain near the ground surface. Therefore, the WBGTs value could increase with increasing moisture content near the urban surface. In addition, $\Delta T_{pop}$ is higher than $\Delta T_{area}$ across all eight regions, which indicates that these adaptation strategies exhibit the potential to reduce not only heat stress but also the pressure of urban crowding under future urbanization.

The implementation of green roofs, cool ground surfaces, and cool walls is projected to reduce $T_{area}$ by 0.27 ± 0.12, 0.24 ± 0.12, and 0.25 ± 0.10 °C, respectively. These findings suggest that roof and ground surface strategies target the average heat stress more directly than do wall strategies. Regionally, building-focused strategies (green roofs or cool walls) are projected to achieve the strongest regional cooling in the Yangtze River Delta (YRD) and Pearl River Delta (PRD). As these zones are high-density urban zones with high roof cover levels and large wall areas, strategies aimed at building surfaces are more effective than strategies aimed at ground surfaces. However, in northern cities with drier

climatic conditions, such as Zhongyuan (ZY), the cool ground strategy achieves a high cooling efficiency because the weaker atmospheric counter radiation conditions in these regions allow more solar radiation to be reflected from the ground back to the upper atmosphere. In addition, compared to the individual strategies, the full adaptation strategy yields more obvious cooling, from 0.2 to 0.4 °C, across all areas, demonstrating particular effectiveness in larger urban agglomerations, where local cooling is projected to exceed 0.40 °C.

**Changes in labor productivity due to future urban warming.** Productivity calculations indicate that future urban warming could lead to substantial labor losses (Fig. 3). Regionally, a significant reduction in worker productivity is projected in densely populated areas along the eastern coast to the MYR under both climate change scenarios, especially in regard to high-intensity

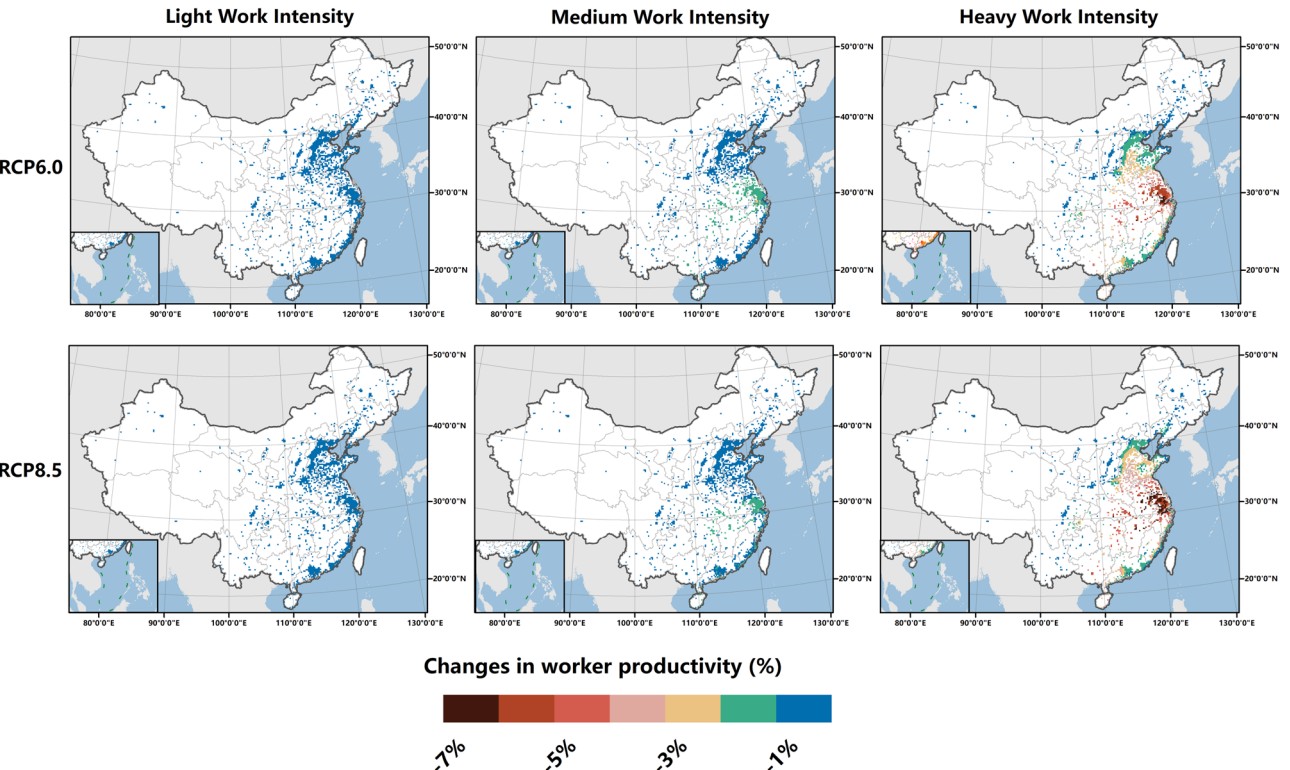

**Fig. 3 Average reduction in the hourly worker productivity for three types of work activities during summertime by the 2050s relative to the 2010s.**
We use the exposure–response function developed by Kjellstrom[11] to estimate the heat impacts on the hourly labor productivity during all possible work hours in summer. For the calculation process, please refer to the Methods section.

work in the YRD. Under rapidly increasing heat exposure and work intensity levels, labor losses relative to the baseline level could exceed 7% in the YRD by the 2050s.

Based on the economic evaluation results, the market costs due to future warming-induced labor losses are projected to reach 5.11 ± 0.49 billion US dollars (RCP 6.0) and 5.82 ± 0.55 billion US dollars (RCP 8.5) per year in all urban grids by the 2050s, more than double the value in the 2010s (2.11 ± 0.35 billion as the baseline level). These economic losses account for 0.20 and 0.25%, respectively, of the yearly total account gross domestic product (GDP) resulting from all exposed populations (please refer to the Methods section for details). In comparison, the total account GDP is consistent with recent results obtained for East Asia[17]. The total projected labor losses are lower than those determined in other nationwide studies[18]; however, we only considered the values in the selected urban grids and did not consider all populations and industry types.

In addition, the total increase in economic costs varies substantially across regions. The cost variations in low-latitude cities, such as cities in the PRD, are negligible due to smaller WGBTs increases and higher AC coverage levels in these cities. However, cities located in mid- to high-latitude areas, such as cities in the MYR, YRD, and Jing-Jin-Ji region (JJJ) and on the Shandong Peninsula (SD), are projected to experience considerable economic losses accounting for more than 0.6% of the total GDP. These high economic costs can mainly be explained by the higher growth in heat exposure and lower AC coverage in these cities than those in other cities.

Among the different sectors, the construction sector will be the most adversely impacted by future urban warming, as this sector requires higher-intensity work and more work activities outdoors. As shown in Fig. 4A, more than 70% of the total economic cost is sourced from this sector. The manufacturing and transportation

sectors, with lower-intensity work and mainly indoor labor activities, are less affected than the construction sector. The service sector exhibits the lowest risk, and the economic cost of the labor loss due to future warming originating from the service sector reaches only 0.18 billion per year relative to the baseline level, as this sector is associated with the lowest work intensity and the most indoor work activities.

Therefore, despite the varied warming trends, as the local economic structure determines the employment patterns of urban residents in the different sectors, the economic structure across cities could impact the future cost. As shown in Fig. 4A and Supplementary Table 1, the differences between the relative contributions of the different sectors to the market cost are notable among the eight urban agglomerations, although the construction sector is a major contributor to the total economic cost across these eight urban agglomerations. In addition, in SD, as a large share of workers is employed in the manufacturing and transportation sectors, the share of the economic cost sourced from the manufacturing and transportation sectors in this region is significantly larger than that in the other regions.

**Avoided labor loss due to adaptation strategies.** Under the RCP 6.0 scenario with the adoption of mitigation strategies, the cooling effect due to the implementation of adaptation strategies prevents some of the labor loss attributed to future urban warming (Supplementary Fig. 3), especially the productivity of high-intensity work in the YRD and PRD.

Regarding the economic effect, the results confirm that cooling due to the implementation of urban adaptation strategies is important for urban economic development. The adoption of cool walls yields more economic benefits than those yielded by the adoption of the other strategies, saving 0.26 ± 0.08 billion

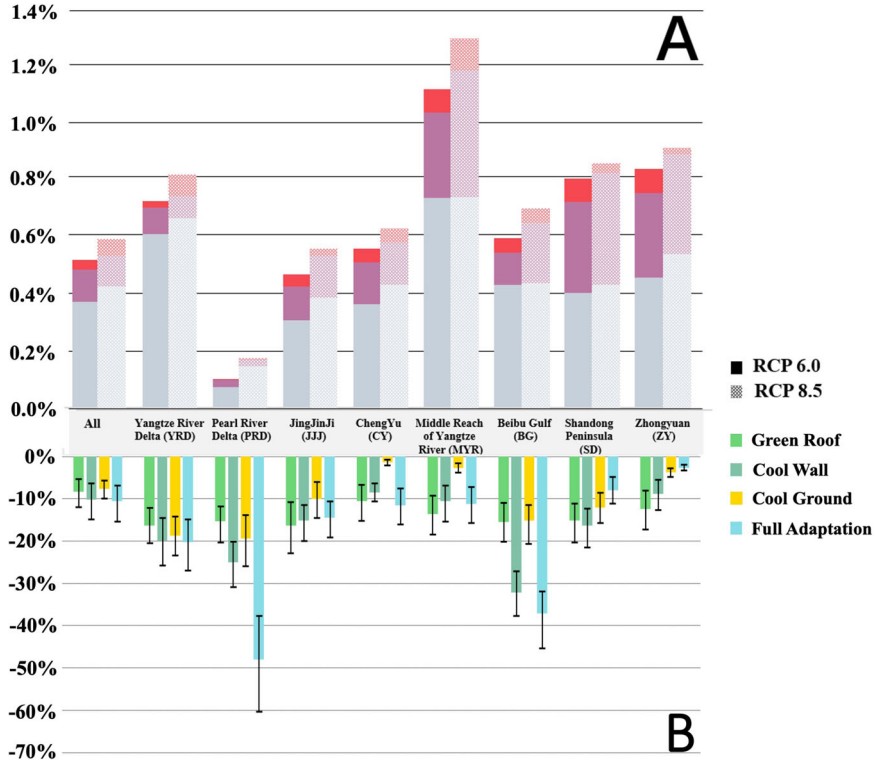

**Fig. 4 Changes in the total GDP loss compared to the baseline scenario and the fraction of the economic losses that can be prevented via the implementation of adaptation strategies in all urban grids.** The percentages in the upper panel (**A**) indicate the percentage increases in the economic losses due to urban warming under the RCP 6.0 and RCP 8.5 scenarios with SSP2 by the 2050s relative to the 2010s, and the three colors indicate the total contribution of the three sectors. The percentages in the lower panel (**B**) indicate the annual percentage reductions in the economic losses due to the different urban adaptation strategies (2050–2060) under RCP 6.0 compared to the additional increases in the economic losses in the future relative to the baseline level. Error bar refers to the standard deviation of the results from 2050 to 2060.

dollars per year based on the lost workforce due to urban warming in the future. Although the cooling effect of green roofs and cool ground surfaces exceeds that of cool walls, the reduction in the total cost due to green roofs (0.21 ± 0.03 billion) or cool ground surfaces (0.19 ± 0.05 billion) is less than that due to cool walls, as shown in Supplementary Table 2. Because walls are distributed within the urban canopy, there are two chances for wall surfaces to occur perpendicular to the angle of solar incidence (morning and afternoon), which extends their cooling duration despite their lower impact on the daily maximum and mean WGBTs values.

At the regional level, the reduction in economic losses in certain lower-latitude regions (such as the Beibu Gulf (BG)) or regions with large urban land-use areas (such as the YRD and PRD) is considerable (Fig. 4B), which can be explained by the limited increases in $T_{area}$ or greater cooling effects due to the implementation of the above adaptation strategies. In addition, the installation of cool walls can recover more economic losses than can the other strategies across all urban agglomerations except the drier agglomerations located in the north (such as JJJ and ZY) and those with limited solar radiation (such as CY and MYR).

Across the different sectors, the trend of the recovered loss magnitude is similar to that of the loss magnitude due to future urban warming, as the construction sector benefits more and the service sector benefits less from outdoor cooling (Supplementary Fig. 3). Therefore, the economic structure across cities affects not only the future warming effect but also the recovery degree due to the different adaptation strategies.

In addition, the spatial variability in $T_{area}$ exhibits a positive relationship with the ratio of economic losses (Fig. 5A). The slope

of the simple linear regression relationship between $T_{area}$ and the ratio of economic losses slightly decreases after the implementation of the above urban adaptation strategies, which suggests that the implementation of these urban adaptation strategies can partially mitigate background climate change but cannot alter the fundamental role of climate change risk. However, as shown in Fig. 5B, resident distribution-induced heat exposure attains no significant relationship with the ratio of economic losses after the implementation of the urban adaptation strategies, especially after full implementation, which indicates that the influence of urban crowding on future labor losses can be expected to decrease after the implementation of these urban adaptation strategies. Therefore, these results highlight that although the absolute impacts of urban adaptation on $T_{area}$ increase or climate change risk are limited, the implementation of the considered urban adaptation strategies yields economic benefits to mitigate the challenges posed by urban crowding.

## Discussion
In this study, the hourly urban labor loss due to future urban warming was explored with a regional climate model coupled with a UCM and an ERF between the WBGTs and labor productivity across 231 major cities in mainland China. The results of the present study facilitated a greater understanding of the regional impact of expected urban warming, and these results were obtained via three approaches: (1) we simulated the heat stress variation in each suburban grid, considering the effects of not only background climate change but also expected regional urbanization. (2) We quantified the economic cost attributed to urban warming, considering the effect of the city-specific

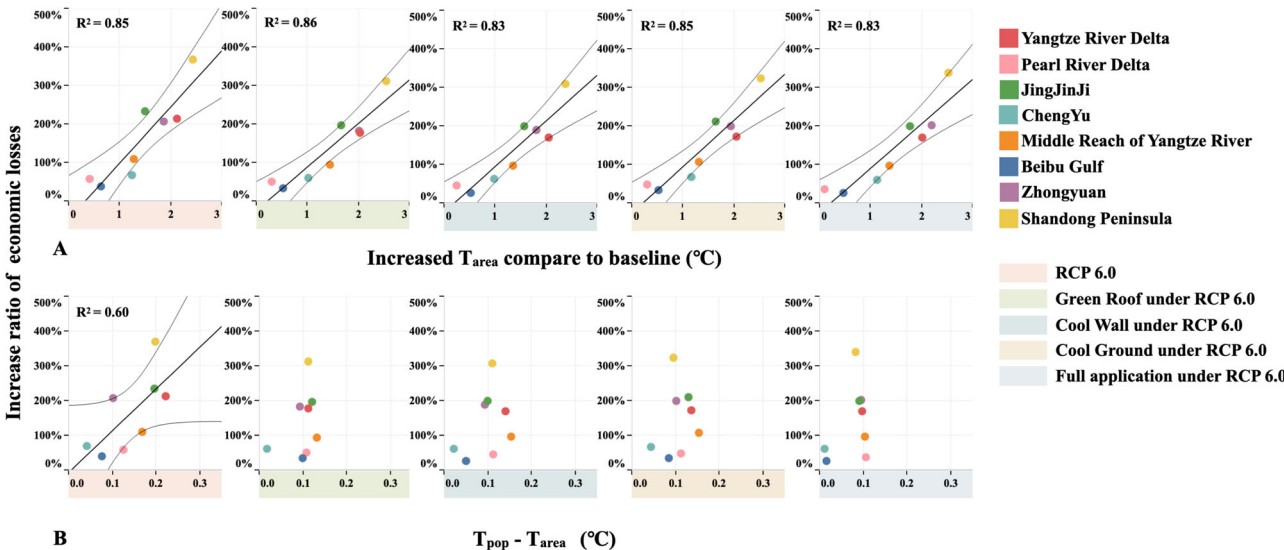

**Fig. 5 Relationships among the increase in the ratio of economic losses, $T_{area}$ increase, and heat exposure increase induced by the resident distribution ($T_{pop}$ − $T_{area}$). A** Relationships between the increase in the ratio of economic losses and $T_{area}$ increase under the different scenarios in individual urban agglomerations. **B** Relationships between the increase in the ratio of economic losses and resident distribution-induced increase in heat exposure. Linear regression is performed with a one standard deviation confidence interval. The bold lines indicate that the linear relationship is statistically significant at the 0.05 level.

economic structure rather than a fixed economic background. (3) We assessed the cost at the suburban scale instead of averaging the cost across whole climatic regions or metropolitan areas.

Our assessment was based mainly on changes in population exposure. To better quantify the impact of expected economic changes under future scenarios, we considered changes in the population distribution to reflect the potential impact of future urbanization in China, where the urbanization rate is projected to increase. In this process, the populations of large cities will continue to increase. Therefore, based on the projected population distribution, we could assess the impacts of future urbanization and local economic development across the different cities.

Previous studies have predominantly focused on the economic consequences of climate change at larger scales, such as entire regions[17,18]. This research indicated the pathways along which climate change generates serious damage in specific sectors within urban areas. Our analysis further revealed that the implementation of suitable adaptation strategies could partially prevent market cost losses. As more people migrate to cities, future urbanization will act as a double-edged sword; urbanization can amplify the average heat exposure level[2] and can distribute the mitigating effects of the implementation of urban adaptation strategies across more urban residents. This situation makes the selection of suitable urban adaptation strategies particularly critical under inevitable future warming conditions. The results of this study provide preliminary guidance to individual cities in the selection of effective climate adaptation measures.

To reflect the cost of applying these adaptation measures, we quantified the cost of the implementation of green roofs in the selected urban areas, as the adoption of green roofs results in the highest cost. For details on the calculation method, please refer to the Appendix. Overall, based on our calculation, approximately 20 years would be required to recover all the application costs. It should be noted that we simplified the quantification calculation here, without considering daily maintenance costs, cost of capital, and time value of money in the recovery process. This simplification should be evaluated with a detailed economic model. Moreover, it has been indicated in many studies that indoor areas can be cooled via the implementation of these building measures,

thereby reducing the total energy use and daily costs in the warmest months to further save costs in the short term[25,26]. Therefore, we believe that the benefits of the application of green roofs exceed the associated costs in the long run.

Based on the results of this study, several critical implications for policy formulation can be identified. First, the findings provide another justification for effective interventions, as most major cities in mainland China are expected to experience significant increases in urban heat stress levels. Second, in revealing that future urban warming disproportionately affects low-income sectors and threatens environmental justice, the results highlight that future urban warming cannot only seriously damage cities as a whole but can also contribute to income inequality, especially in well-developed large urban agglomerations. This inequality-related impact further emphasizes the importance of climate change policies that can prevent work efficiency losses in lower-income and climate-sensitive sectors to achieve future urban fairness. In addition, the results confirm the uneven distribution of urban warming under climate change due to the varying regional intensities of average solar radiation reception and atmospheric inverse radiation. Moreover, the total labor loss attributed to urban warming is projected to differ due to varied local socioeconomic factors. There exists a gap between outdoor cooling and labor productivity recovery levels. Therefore, policy options focused on urban heat adaptation strategies should recognize the relative growth in heat stress and local-scale multisector socioeconomic results rather than proposing a single goal to fit all cities. Moreover, the cooling strategies selected in this study could reduce the average $T_{area}$ value by more than 0.20 °C, while greater average warming of 1–2 °C could persist. Thus, although these strategies could reduce heat pressure, with cooling in excess of 1.5 °C at the peak of solar radiation, a gap remains relative to the average warming level. Therefore, in addition to the adaptation strategies considered in this study, other adaptation strategies, such as shifting working hours in the construction sector via mechanization, should be considered in the future.

Several limitations in this study should be noted. First, our dataset lacked information on how the work income per hour or local economic structure in the different regions is projected to

change under certain SSPs. However, based on baseline economic structure data and projected population distributions, our assessment considered the influences of the local economic structure and future urbanization at the suburban scale. Second, we did not consider how the AC coverage level or working conditions in the different sectors might vary in the future, which might lead to over- or underestimation, especially in regard to the various types of medium-intensity work activities. Thus, we could not precisely estimate the impacts of future economic development other than future population changes and variations in working conditions on urban labor losses. Furthermore, we applied only one type of epidemiological ERF, and no additional questionnaires were employed. The relationship between heat conditions and work loss could vary across the entire region. Therefore, the exposure–response relationship should be further examined. We will address this aspect in future work drawing on multiscenario assumptions supported by a more detailed dataset on a fine scale.

Most cities worldwide are projected to expand in the foreseeable future[27]. Therefore, the level of heat exposure experienced by residents is also expected to increase. The expected significant impacts of this increased heat exposure on human health have been studied in detail, while the impact of outdoor heat on the labor capacity or productivity remains unclear. The present study suggests that heat exposure among urban residents could significantly impact the labor capacity or productivity across cities in China. Under rapid urbanization worldwide, the increased burden of labor loss elsewhere could be as serious as that reported for China. Guaranteeing occupational safety will remain a crucial target of future urban planning. Our study joins the call for a better understanding of local heat exposure changes in the future, and the analysis in this study provides a methodological framework to inform city infrastructure plans.

## Methods

**Scenario design**. We attempted to quantify the risk of summer heat pressure by combining climate modeling over the baseline period (i.e., the recent past [June 1 to August 31 in each year from 2010 to 2020]) with climate projections for the middle of this century ([June 1 to August 31 in each year from 2050 to 2060]) based on climate change models with existing population projections.

Regarding future climate change, as the Chinese government has recently pledged to achieve carbon neutrality by 2060 and has already adopted a portfolio of low-carbon policies to meet China's nationally determined contribution vowed under the Paris Agreement considering the 2 °C global temperature target[22], we expect that carbon emissions in China will be controlled in the near future. Therefore, we selected RCP 6.0 as the future global climate change scenario, which assumes a decreased greenhouse gas emission rate due to technologies and strategies targeting greenhouse gas emission reduction in the future. Regarding urbanization, the percentage of the total population of China living in urban areas is 60.9%. The Chinese government expects this value to increase to 65.5% by 2030[23], and in the 2018 revision of World Urbanization Prospects, the United Nations reported that the percentage of the total population of China living in urban areas is expected to surpass 70% by 2050[24]. According to this background, urbanization in China is expected to maintain its current growth trend in the near future, with no significant acceleration or slowdown. Therefore, we selected shared socioeconomic pathway 2 (SSP2), under which the fertility, mortality, migration, and urbanization development rates in China are all maintained at their current levels. According to research[3], SSP2 is largely compatible with the RCP 6.0 climate scenario in terms of assumptions about adaptation and mitigation policies.

**Climate modeling**. To model future regional-scale climate conditions, we applied the advanced research version of the regional climate Weather Research and Forecasting (WRF) model[28] to downscale RCP 6.0 assumptions across mainland China. The WRF model is a state-of-the-art weather forecasting and climate simulation model for use at the regional scale and has been widely applied in urban climate studies worldwide[1,2]. In regard to the initial boundary conditions under the RCP 6.0 scenario, we used global bias-corrected dataset outputs retrieved from the National Center for Atmospheric Research (NCAR) community earth system model, which participated in Phase 5 of the Coupled Model Intercomparison Project.

In addition, to reflect the characteristics of future urbanization, including the potential urban land-use expansion and crowding under future scenarios, we selected a single-layer UCM to study urban-related climate processes[29]. The UCM can consider building shadows, urban anthropological heat emissions, building reflections, and several other key factors influencing urban weather conditions[30]. We employed several observation and simulation datasets to adjust the preset settings in the UCM. First, urban land-use data for the 2050s under the SSP2 scenario were obtained from related research[31]. Compared to the 2010s, the area of urban land use increased by more than 80,000 km$^2$ across China by the 2050s under the SSP2 scenario. Second, the distribution of urban residents by the 2050s under the SSP2 scenario was obtained from the NCAR Integrated Assessment Modeling Project[27,32]. Compared to the 2010s, the urban population increased by 255.48 million by the 2050s under the SSP2 scenario. We updated the land-use settings of the WRF model based on these urban land-use and population data (please refer to the Appendix for details) and combined it with climate change scenarios in the simulation of WBGT changes.

We considered all urban areas across mainland China as the research area. In addition, in the 14th Five-Year Plan, the Chinese government highlighted the development of 19 urban agglomerations in the future. Considering the varied socioeconomic and climatic conditions of these agglomerations, we selected eight urban agglomerations with a high heat pressure during the summer as key analysis units (Supplementary Fig. 1). According to our results, more than 90% of the total high heat stress-induced labor loss originates from these areas. Therefore, we should give more attention to these urban agglomerations to address the problem of future urban warming.

For additional important information on physical attribute options, model operation, and model calibration, please refer to the Supplementary Methods.

**Adaptation strategies**. A city exhibits a high concentration of buildings, and the interface between manmade surfaces and the urban atmosphere comprises three types: roofs, walls, and ground surfaces (or pavement). Several mitigation strategies for urban heat reduction have been proposed[1], most of which involve cooling through the modification of the materials of these three interfaces to reduce the sensible heat available for transmission into the air or toward other envelopes.

Reflective-coating and greening technologies are immensely useful for improving the thermal performance of existing building envelopes at a low cost[33]. Regarding roofs, many cities in China, such as Shanghai, Nanjing, Guangzhou, and Shenzhen, have introduced a series of policies to encourage green-roof installation[34,35]. Specifically, the Shanghai Municipal Greening Regulations require all newly built public buildings to adopt green roofs, and the total green-roof coverage is required to be not lower than 50%. In Hangzhou, over 420,000 m$^2$ of green roofs has been added since 2010. Based on this background, cities in China are expected to exhibit high green-roof coverage levels in the future. Therefore, under the green-roof scenario, we considered vegetation cover options with an adequate irrigation supply that could be applied to 80% of the roofs in urban areas. In regard to walls and ground surfaces, high-reflection technologies are the most widely used cooling technologies[24,36]. Although there exist certain gaps between the current green-roof coverage state and our modeling settings, our goal was to demonstrate the potential impact of a feasible near-future scenario involving the upper bound of climate effects based on the status quo and a relevant study[1]. In addition, the simulated green-roof type is one of the most common types of green-vegetation roof coverings and comprises four layers (from bottom to top): a common concrete roof layer, a growing medium layer, a loam soil layer to support grass, and a grassland layer[35]. The cool wall (or reflective wall) scenario assumed uniform application of reflective-coating materials with an albedo value of 0.80 on all wall surfaces. This albedo value represents an increase of 0.6 over the baseline wall albedo values, and these values were sourced from previous research[30,37]. Similarly, the cool ground scenario assumed uniform application of a 0.6-albedo coating material to all urban ground or pavement surfaces. According to an observational study in China[38], several ground materials with high reflectivity attributes can achieve reflection within this range. An albedo value of 0.6 indicates an increase of 0.4 over the baseline ground albedo value according to field observations[39]. The full adaptation scenario included the combination of green-roof, cool wall, and cool ground strategies.

Although these scenarios were not based on specific economic or policy projections and currently available urban cooling measures are applied infrequently and at low intensity, we established these adaptation scenarios as representative potential policies that may be implemented in the future. Via fully coupled and process-based modeling, this study revealed the sensitivity of urban heat conditions to effective cooling interventions and could thus inform the development of climate-related plans aimed at reducing urban heat through infrastructure-related intervention strategies.

**Quantifying the change in labor productivity due to urban heat**. To quantify the changes in labor productivity due to urban warming, we used the ERF capturing the relationship between heat pressure and labor productivity developed by Kjellstrom[11] and the High Occupational Temperature Health and Productivity Suppression program[40]. This ERF describes the correlation between the labor

productivity and WBGT as follows:

$$\text{Workability} = 0.1 + \frac{0.9}{\left(1 + \left(\frac{WBGT}{w1}\right)^{w2}\right)} \qquad (1)$$

where $w1$ and $w2$ are the specific WGBT values derived under each of the three work intensities: low (24.64 and 22.72, respectively), medium (32.98 and 17.81, respectively), and high (30.94 and 16.64, respectively). The relationship between the WGBT and ratio of labor losses under the three work intensities is shown in Supplementary Fig. 4.

This function has been widely adopted in studies such as that of Chavaillaz[41], who used this function to assess the effect of the global heat pressure on labor loss associated with global $CO_2$ emissions. Knittel[42] used this function to assess labor losses under certain climate and population projections. Orlov[17] employed this function to calculate the market costs of future global warming-induced reductions in labor productivity worldwide.

In this study, before using this function, we calculated the WBGT value in each urban grid. We calculated the WBGT under the different scenarios by using the method of Bernard and Pourmoghani[43]. For details on the calculation method of the WBGT, please refer to the Supplementary Methods.

We only considered urban residents of the population, as one of our aims was to assess the potential influence of urban adaptation strategies. We included 16 types of sectors most likely employing urban residents. Based on a previous study[17,18], we divided these sectors into three types according to the work intensity. Low-work intensity sectors included trade and all retail sales, hotels and restaurants, telecom and software services, financial services, residential real estate, leasing and business services, scientific research and technical services, public administration, education, recreational and service activities, social security, and insurance. The medium-work intensity sectors included transportation and most manufacturing sectors, such as machinery, electronic equipment, and other industries. The high-work intensity sector included only the construction sector. Other types of sectors not employing urban residents, such as the agriculture, forestry, fishery, and extraction sectors, were excluded from analysis in this study.

Next, we calculated the total number of residents in each urban grid of city $i$ employed in sector $i$ as $Pop_i^j$:

$$Pop_i^j = Pop^j \times R_{\text{adults}}^j \times \left(1 - R_{\text{unemploy}}^j\right) \times R_i^j \qquad (2)$$

where $Pop^j$ denotes the number of total urban residents in the urban grid of city $j$, $R_{\text{adults}}^j$ and $R_{\text{unemploy}}^j$ denote the ratio of the working population to the total population and the ratio of the unemployed population to the total population of city $j$, respectively, and $R_i^j$ denotes the proportion of the working population employed in sector $i$ in city $j$.

The AC system is an effective cooling adaptation strategy to prevent extreme heat effects. Based on a previous study[17], we assumed that the work productivity is unaffected by urban warming when AC systems are available. Therefore, we excluded those workers protected by AC systems:

$$CPop_i^j = Pop_i^j \times \left(1 - C^j\right) \qquad (3)$$

where $CPop_i^j$ denotes the total number of residents in each urban grid of city $j$ employed in sector $i$ who are not protected by AC systems, and $C^j$ denotes the penetration rate of AC systems in city $j$ for all sectors except the construction sector, as the latter sector mainly entails outdoor work.

We next calculated the hourly work loss ($WL$) in each sector $i$ in each urban grid of city $j$ as:

$$WL_{ij} = CPop_i^j \times (1 - \text{Workability}) \times W_i^j \qquad (4)$$

where $W_i^j$ denotes the hourly wages in sector $i$ in city $j$.

Data on city-specific ratios of the average working population and unemployed population, proportion of the working population employed in each sector, and hourly wages in each sector were obtained from the China City Statistical Yearbook (2010–2019). The penetration rate of AC systems was calculated based on data on the ownership of AC systems per 100 households for each province, which was obtained from the China Statistical Yearbook (2010–2020). We assumed that AC system ownership in each city was consistent with the average level in the province to which the city belonged. The workplace penetration rate of AC systems could differ from the household ownership rate. Due to a lack of data, we assumed that the penetration rate of AC systems in each sector except the construction sector followed the same trend.

Theoretically, the maximum working hours for a working person in summer was calculated as three summer months times 30 days per month times 12 h per day (from 8:00 am to 20:00 pm). We then determined the annual total labor loss due to heat during summer in each urban grid.

Overall, this study involved 231 cities out of a total of 333 Chinese cities and 196 million urban residents out of a total of 643 million urban residents. These cities were selected based on the criterion of an urban land area above a certain size (10 km²). We also calculated the yearly total GDP in each selected urban grid for comparison. The yearly total GDP was obtained in two steps: (1) the account GDP of each city was calculated by multiplying the total population within all selected urban grids by the GDP per capita of the corresponding city. (2) The GDP values of each city were summed to obtain the total account GDP.

All these calculations were performed on the Python platform using the open-source *pandas* and *NumPy* libraries.

**Reporting summary.** Further information on research design is available in the Nature Research Reporting Summary linked to this article.

## Data availability
All the initial boundary conditions data used in this study can be downloaded from The Research Data Archive managed by the National Center for Atmospheric Research at https://rda.ucar.edu/. Other relevant data in this study are available from the corresponding authors upon reasonable request.

## Code availability
The WRF model source code, documentation, and other resources can be found at http://www2.mmm.ucar.edu/wrf/users/.

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

## Acknowledgements

This work was conducted when C.H. visited Boston University in 2019–2020, which was supported by the China Scholarship Council (grant no. 201906100127). H.K. was supported by the National Natural Science Foundation of China (grants 92043301 and 82030103). A.S. was supported by the European Union's Horizon 2020 Project EXHAUSTION (grant agreement No. 820655).

## Author contributions

C.H. devised the project and conducted the interpretation and analyses, and wrote the original draft; W.M. revised part of the manuscript, and jointly discussed the interpretation and results with P.L.K. and H.K.; Y.Z. edited the revised manuscript; A.S., R.C., and Y.Z. participated in stages of paper revisions.

## Competing interests

The authors declare no competing interests.
