## [Peer Review File · Nature Communications]

The inequality labor loss risk from future urban warming and adaptation strategiesREVIEWER COMMENTS

Reviewer #1 (Remarks to the Author):

In this study, the authors find out inequality labor loss risk, the economic cost of urban warming, cost at a sub-urban scale and the author is highly appreciated to focus on a key issue that is rarely studied. However, in its current form, the paper has some major shortcomings. Language editing is a must and the grammar and punctuations need to be edited for corrections throughout the manuscript. The manuscript requires major revisions with queries indicated that need to be addressed. The manuscript could make a worthy addition to the existing knowledge if the manuscript is revamped and written clearly, as it now lacks clarity and focuses on few areas.

Abstract:

- Mention the type of study and add study design and study period, method of data collection.
- Add more points on the methodology of the study so that it can be replicated by other researchers for their countries.

Introduction:

- Even though the content is well organized, there are few concerns.
- Add more on how heat affects the workers and its impacts on health and work efficiency with the supporting literature which may strengthen the article.
- Try to add more references in this section.
- Change the reference formatting in lines 64 and 65.
- Citation of the supplementary file is not necessary in the introduction section, e.g “Eight urban agglomerations (Fig.S1) with the differentiated socio-economic”

Results:

- Overall the results were well written but all the paragraphs, start with the description of all the tables, figures, describe your study results, refer to the table, and figure names accordingly. Try to modify slightly to avoid redundancy as it is already apparent.
- Mention the future CC Projection model name in the methodology section, even though it has been mentioned in the supplementary file.
- The language part has to be corrected throughout the manuscript. Use free English editing software, which is available online. For e.g “Grammarly” to edit the language part.
- Line 89, 154 “strategies could reduce the average Tarea by over 0.20 °C in most of the 8 urban agglomerations” give spacing to “Tarea”. Follow the uniformity throughout the manuscript.
- Reference number 21 is in line 119 but reference number 20 is in line 97. Be cautious while citing the supporting articles.
- As mentioned in the abstract “We combine hourly heat stress data with exposure-response functions between heat exposure and labor productivity to answer these questions for different cities in China under future climate change”. I presume that you had used the formula to see the exposure-response functions. Were any questionnaires used to collect this data? Kindly clarify.
- Mention the type of statistical tool used for the analysis.
- The citations in this manuscript should be in superscript format. Try to change all the references accordingly as per the guidelines.
- Add full forms in the table caption rather than the abbreviations.
- Figure 1: try to represent the temperature variation in the map which has boundaries/lines for each province mentioned in the article rather than the national boundary. So that the reader can interpret the results easily.
- How the work intensity has been classified, mention the standard for that.
- Add the note for figure 4 and include the full form for all locations.

Discussion:

- Try to add supporting literature for the phrases which support your study results.
- Mention the full form of SSPs in line 208 as it is the first phrase that has an abbreviation (SSPs).

Methods:

- Describe study design and approach.
- Describe the inclusion and exclusion criteria of the study, if any.
- Capitalize the urban canopy model (UCM)
- Add about the ethical approval, if any as per your country guidelines

Conclusion:

- Add conclusion for your study

Author contribution:

- Add the Author's contribution to the study.

Acknowledgment:

- Add the Acknowledgement for the study.

Competing interest:

- Add the competing interest for the study.

References:

- Try to follow the author's instruction.

Reviewer #2 (Remarks to the Author):

Main comment:

The manuscript estimates the heat-induced reductions in labor capacity for urban areas in China under future climate change using a regional climate model. Only one climate change scenario was considered using RCP 6.0 and SSP2

The main finding is that heat stress will lead to an estimated annual economic loss of 3 billion USD, with a confidence interval of 0.14. The study does not compare its absolute figures in relation to the overall economy (or the share of the urban economy in China's GDP), which would reveal that the estimate damage is just 0.02% of China's current GDP (2019). This seems far too low and a discussion of the relative magnitude of the result is unfortunately missing – but is definitely needed as other studies (e.g. Orlov et al. 2020 in *Global Env. Change*) report much higher magnitudes in GDP declines. The authors compare the results once with a conference paper from Szewczyk et al. (2019) (the reference is incomplete, by the way), however that study also reports much higher magnitudes. Also, the potential of urban adaptation strategies, which are assessed in the study, need to be put into perspective. The study reports that they “could offset more than 0.15 billion” (I assume USD, there was no unit reported). The study again does not discuss the relative magnitude of this major result except for Fig 4b. The adaptation strategies investigated could only offset 5% (or 10% as the 0.15 billion reported in the abstract contradicts Table S2 and Fig 4b) of the economic damage, despite the highly ambitious implementation rates. For instance, the green roof scenario assumes an adaptation rate of 80% across mainland China and the cool-wall scenarios assumes a reflective coating for all wall surfaces in urban areas. The scope and ambition of these scenarios are doubtful, and still produce only a very small impact. Why?

As a reader, one is left wondering why the authors did not focus on this part of the story:

1) Heat stress in Chinese major urban areas causes “just” 3 billion USD in annual economic losses derived from reductions in labor capacity

2) Very ambitious adaptation strategies only help to reduce this loss in worker productivity by 5 (or 10%)

A possible or even likely reason is that the authors did not reflect about their results this way. If true, this substantially erodes my trust in how rigidly and carefully the analysis has been conducted and interpreted.

There are major results that are unrecognized (or unreflected), which seem to be in conflict with literature and which may point to a flaw in the data analysis and interpretation. Also, the results are taken from just one climate change scenario, without a sensitivity analysis and the adaptation strategy scenarios are highly unrealistic. Moreover, crucial aspects are not covered by the analysis. What is the cost-effectiveness of the adaptation measures? The authors seem to implicitly assume no cost, but that is of course not the case. How will technology and the economy react towards increases in heat-stress? What is the sensitivity of results coming from the model's and scenario's parameters and assumptions?

I therefore doubt that this is a rigid analysis, which probably requires more than a major revision.

General remarks:

Please use a professional English language editing service. The manuscript is difficult to understand in its content, which to a large part is owed to poor language and syntax. The reader is often left guessing what the authors' intended to say. This makes it quite difficult to understand the details of the analysis. There are also many typos, which can be easily avoided using spell check

Remaining major comments:

1. The study's main contribution, beside its empirical findings, is on how in detail the economic structure and population residence is accounted for when analyzing the labor induced heat stress. This is indeed a valuable contribution, but quite technical and not well highlighted. It also does not seem to have a major impact on the results (the % bias it corrects is not explicitly reported, it seems to be less than 10%).
2. China may certainly be a very exciting case study, but the reader does not really see why. This could be much better motivated at the beginning (e.g., X% of the world's urban population live in China)
3. The study does not discuss or consider how cost-effective the adaptation strategies could be. This is a major shortcoming, but could be rectified.
4. I appreciate that the authors try to understand how heat-stress impacts different economic activities at the (sub-)urban level instead of the regional or even national scale. Yet, it remains unclear how the authors estimated the location of the economic activities and how exactly they have treated the seasonality of work loads (e.g., construction workers may not work during the midday heat). Also, such a contribution (while definitely valuable) may not be sufficient for a consideration in a top multi-disciplinary journal. This is also backed by the rather simplistic consideration of other dimensions (economic adjustments are absent, reliance on one SSP, RCP and exposure function, etc).
5. The economic estimation is very simple and static. There is no endogenous adaptive behavior. Also, no changes in the intensity of the individual economic activities (or even technological improvement). However, the Chinese economy is growing very rapidly and technologies are evolving, so is it reasonable to still assume the same economic structure of today?
6. The discussion of how the analysis accounts for changes in the urban population are very difficult to comprehend. I think the authors at least once confuse "urbanization rate" with "share of total population living in urban areas" – see line 236.

Minor comments:

1. Defining transportation, manufacturing, etc. as medium work-intensity may overestimate heat-stress (also many of these activities can or are done in AC environments - especially until 2050)
2. I think the term "labor support" is not helpful for an interdisciplinary audience. I would also recommend to speak of "losses in labor capacity or productivity" than just "labor loss".
3. There are many typos, unclear sentences, strange formatting, etc. - this gives the reader a sloppy impression.

Reviewer #3 (Remarks to the Author):

This study assesses the heat-induced impacts on labour productivity and associated economic costs under different adaptation measures at a sub-urban scale. The results from this impact assessment provide some novel and useful insights. Especially, I find interesting the combination of a regional climate model with an urban canopy model to investigate the effectiveness of different adaptation options in urban areas. I have a few questions and improvement suggestions:

- 1) In Lines 205-206, the authors write: "First, we established only one future climate-society scenario, the modelling results would be slightly different if we compared the results from scenarios under different assumptions". I am not sure that the results would be 'slightly' different. The uncertainties associated with climate modelling (i.e., climate sensitivity) are large, so are socio-economic uncertainties. I guess the authors did not have enough capacity to conduct a dynamic downscaling for multiple RCPs and climate models. However, I wonder if it makes sense to consider, two or three SSPs to investigate the relevance of socio-economic uncertainties.
- 2) I am a bit surprised that the economic cost from construction is so large compared to other sectors. I understand that construction is more labour-intensive and requires working outdoors, but I guess much more people are working in manufacturing and services. So, I would expect that the total cost of heat-induced impacts on labour productivity in manufacturing and services would be larger than in construction. On that regard, I think it would be useful to have a table or figure showing the sectoral wage rates and the number of people employed in each sector and region. Also, please report the recent penetration of air conditioners by regions.
- 3) I would expect that the economic structure will change over time (i.e., more income will be generated from the service sector compared to manufacturing and construction). However, it seems that a structural change in the regional economies is not implemented in this study. In this case, this should be mentioned as a limitation.
- 4) Also, there are other adaptation measures, which could be relevant to urban areas. For example, mechanization in construction, shifting working hours, acclimatisation etc. The authors should mention this in the discussion.
- 5) Regarding the presentation of economic impacts, I find it useful to show not only the total cost but also relative impacts, such as the cost per worker (e.g., relative changes in income per worker or USD lost per worker due to heat stress). One could also show the cost per worker in comparison to the income per worker in different regions to reveal potential impacts on income inequality and distribution across regions.
- 6) Fig. 4 shows that in some regions, the annual economic loss could be larger than 200%, while the labour productivity loss does not exceed 8%. The authors should better explain the calculation of economic losses. The calculated economic costs seem to be implausible large.
- 7) As far as I know, the Bernard & Pourmoghani's method requires an iterative solution, but the method for WBGT calculation, which is described in Supplementary Information, does not seem like an iterative method. Please clarify it. Do you use an original Bernard & Pourmoghani's method or its simplification? Also, please add the formula for calculation of the dewpoint temperature.
- 8) I would suggest that the manuscript should undergo extensive English revisions because there are many typos and errors.

Responses to the reviewers' comments

Reviewer #1:

In this study, the authors find out inequality labor loss risk, the economic cost of urban warming, cost at a sub-urban scale and the author is highly appreciated to focus on a key issue that is rarely studied. However, in its current form, the paper has some major shortcomings. Language editing is a must and the grammar and punctuations need to be edited for corrections throughout the manuscript. The manuscript requires major revisions with queries indicated that need to be addressed. The manuscript could make a worthy addition to the existing knowledge if the manuscript is revamped and written clearly, as it now lacks clarity and focuses on few areas.

1. Abstract: Mention the type of study and add study design and study period, method of data collection. Add more points on the methodology of the study so that it can be replicated by other researchers for their countries.

Response:

Information on the study design, study period and methods of data collection has been added to the abstract as basic research background. Please see the new abstract below:

“We combine hourly heat stress data modeled with a regional climate model with exposure-response functions between heat exposure and labor productivity obtained from related studies to investigate this possibility for 231 Chinese cities. We find that future urban heat stress is projected to cause an increase in labor losses of more than 0.20% total account GDP per year in the 2050s relative to the 2010s, with the largest increase projected to occur in mid-latitude coastal areas, and that some lower-paid sectors are disproportionately impacted.”

2. Introduction: Even though the content is well organized, there are few concerns. Add more on how heat affects the workers and its impacts on health and work efficiency with the supporting literature which may strengthen the article. Try to add more references in this section.

Response:

Four additional references have been cited in the main text regarding the effects of heat on health and work efficiency (L42-43), including:

11) Kjellstrom Tord, Freyberg Chris, Lemke Bruno, Otto Matthias, Briggs David. Estimating population heat exposure and impacts on working people in conjunction with climate change. *International Journal of Biometeorology* 62, 291-306 (2018).

12) Orlov Anton, Sillmann Jana, Aaheim Asbjørn, Aunan Kristin, De Bruin Karianne. Economic losses of heat-induced reductions in outdoor worker productivity: a case study of Europe. *Economics of Disasters and Climate Change* 3, 191-211 (2019).

13) Xiang Jianjun, Bi Peng, Pisaniello Dino, Hansen Alana. Health impacts of workplace heat exposure: an epidemiological review. *Industrial health* 52, 91-101 (2014).

14) Gao Chuansi, Kuklane Kalev, Östergren Per-Olof, Kjellstrom Tord. Occupational heat stress assessment and protective strategies in the context of climate change. *International Journal of Biometeorology* 62, 359-371 (2018).

3. Change the reference formatting in lines 64 and 65.

Response:

The references were prepared using Endnote, and the reference formats have been checked and revised as appropriate throughout the manuscript.

4. Citation of the supplementary file is not necessary in the introduction section, e.g “Eight urban agglomerations (Fig.S1) with the differentiated socio-economic”

Response:

The citation has been revised according to the suggestion. L71-73 of the Introduction section have been revised. Please see the new contents below:

“This study covers 18 types of sectors from 231 cities across mainland China. Eight urban agglomerations with different socioeconomic and climatic backgrounds are selected as the key analysis units.”

5. Results: Overall the results were well written but all the paragraphs, start with the description of all the tables, figures, describe your study results, refer to the table, and figure names accordingly. Try to modify slightly to avoid redundancy as it is already apparent.

Response:

We have avoided redundancy according to the suggestion. In the Conclusion section, L78, 132, 143, 167 and 205 have been revised to avoid the start with the description of the tables or figures.

6. *Mention the future CC Projection model name in the methodology section, even though it has been mentioned in the supplementary file.*

Response:

The projection model name has been added accordingly. The methodology section has been revised (L341-344). Please see the new contents below:

“For the initial boundary conditions under the RCP 6.0 scenario, we used the global bias-corrected dataset outputs from the National Center for Atmospheric Research (NCAR) community earth system model, which was used in phase 5 of the Coupled Model Intercomparison Project (CMIP5).”

7. *The language part has to be corrected throughout the manuscript. Use free English editing software, which is available online. For e.g “Grammarly” to edit the language part.*

Response:

In addition to using “Grammarly”, we have had the revised manuscript edited for English language by American Journal Experts.

8. *Line 89, 154 “strategies could reduce the average Tarea by over 0.20 °C in most of the 8 urban agglomerations” give spacing to “Tarea”. Follow the uniformity throughout the manuscript.*

Response:

The nonuniformity problem has been corrected here and throughout the manuscript; please see L102.

9. *Reference number 21 is in line 119 but reference number 20 is in line 97. Be cautious while citing the supporting articles.*

Response:

The references were prepared using Endnote, and the reference formats have been checked and revised as appropriate throughout the manuscript.

10. *As mentioned in the abstract “We combine hourly heat stress data with exposure-response functions between heat exposure and labor productivity to answer these questions for different cities in China under future climate change”. I presume that you had used the formula to see the exposure-response functions. Were any questionnaires used to collect this data? Kindly clarify.*

Response:

In the revised version, we have described the use of only one type of epidemiological exposure-response function without additional questionnaires as one of the limitations of the study; please see L284-286.

“we applied one type of epidemiological ERF, employing no additional questionnaires. The relationship between heat conditions and work loss could vary across the region. Therefore, more studies about the exposure-response relationship are needed.”

11. Mention the type of statistical tool used for the analysis.

Response:

All of the calculations were performed on the Python platform using the open-source libraries *pandas* and *NumPy*. This information has been added to the Methods section. Please see L452 to 453.

12. The citations in this manuscript should be in superscript format. Try to change all the references accordingly as per the guidelines.

Response:

As noted above, the references were prepared using Endnote, and the reference formats have been checked and revised as appropriate throughout the manuscript.

13. Add full forms in the table caption rather than the abbreviations

Response:

The full forms have been added to the table captions accordingly.

14. Figure 1: try to represent the temperature variation in the map which has boundaries/lines for each province mentioned in the article rather than the national boundary. So that the reader can interpret the results easily.

Response:

The province boundaries have been added to Figure 1 accordingly.

Please see the new Figure.1 below:

15. How the work intensity has been classified, mention the standard for that.

Response:

The work intensity was classified based on the following studies:

1. Zhang Yuqiang, Shindell Drew T. Costs from labor losses due to extreme heat in the USA attributable to climate change. *Climatic Change* 164, 1-18 (2021).

2. Orlov Anton, Sillmann Jana, Aunan Kristin, Kjellstrom Tord, Aaheim Asbjørn. Economic costs of heat-induced reductions in worker productivity due to global warming. *Global Environmental Change* 63, 102087 (2020).

To clarify this, we have cited these two references in the Methods section; please see L409.

16. Add the note for figure 4 and include the full form for all locations.

Response:

The full name of all locations has been added to Figure 4.

17. Discussion: Try to add supporting literature for the phrases which support your study results.

Response:

We have cited a series of literature in L90, 104, and 148 to support our results. The added literature including:

12) Liu Xingcai. Reductions in labor capacity from intensified heat stress in China under future climate change. *International Journal of Environmental Research and Public Health* 17, 1278 (2020).

23) Krayenhoff E Scott, Moustauoui Mohamed, Broadbent Ashley M, Gupta Vishesh, Georgescu Matei. Diurnal interaction between urban expansion, climate change and adaptation in US cities. *Nature Climate Change* 8, 1097 (2018).

24) Georgescu Matei, Morefield Philip E, Bierwagen Britta G, Weaver Christopher P. Urban adaptation can roll back warming of emerging megapolitan regions. *Proceedings of the National Academy of Sciences* 111, 2909-2914 (2014).

18. Mention the full form of SSPs in line 208 as it is the first phrase that has an abbreviation (SSPs).

Response:

We have defined SSPs after its first appearance in the manuscript.

19. Methods: Describe study design and approach; Describe the inclusion and exclusion criteria of the study, if any; Capitalize the urban canopy model (UCM); Add about the ethical approval, if any as per your country guidelines.

Response:

We added information about the study design and approach at the beginning of the Methods section. Please see L282 to L285. “By using a regional climate model, we aimed to quantify the potential influences of the interactions among climate change and potential urban adaptation strategies (installation of green roofs, cool walls, and cool ground surfaces) on future urban warming. Based on a widely used ERF^{17, 18}, we assessed the economic costs to urban residents induced by urban heat.”

Regarding the inclusion and exclusion criteria of the selected cities, the chosen cities were selected because their urban land area is larger than a certain size (10 square kilometers). We have added this information to the Methods section; please see L446-447. “The study involved a total of 18 sectors in 231 cities across mainland China. The cities were selected based on the criterion of an urban land area above a certain size (10 square kilometers).”

The term 'urban canopy model' has been capitalized on L314.

20. Conclusion: Add conclusion for your study; Author contribution: Add the Author's contribution to the study. Acknowledgment: Add the Acknowledgement for the study. Competing interest: Add the competing interest for the study. References: Try to follow the author's instruction.

Response:

All of this information was included in the cover letter, and we cannot show this information due to the double-blind review process.

Reviewer #2:

Main comment:

1. The manuscript estimates the heat-induced reductions in labor capacity for urban areas in China under future climate change using a regional climate model. Only one climate change scenario was considered using RCP 6.0 and SSP2.

Response:

We thank the reviewer's comments. Base on the reviewer's suggestion, we have added another scenario, RCP 8.5 in the revised manuscript, and have compared our results with our previous results to make our results and discussion more reliable.

We have also compared the labor and economic losses under the two different climate change scenarios. The results show that the economic losses under the two different scenarios are comparable but that the spatial pattern differs between these two scenarios due to the different warming distributions.

2. The main finding is that heat stress will lead to an estimated annual economic loss of 3 billion USD, with a confidence interval of 0.14. The study does not compare its absolute figures in relation to the overall economy (or the share of the urban economy in China's GDP), which would reveal that the estimate damage is just 0.02% of China's current GDP (2019). This seems far too low and a discussion of the relative magnitude of the result is unfortunately missing – but is definitely needed as other studies (e.g. Orlov et al. 2020 in Global Env. Change) report much higher magnitudes in GDP declines. The authors compare the results once with a conference paper from Szewczyk et al. (2019) (the reference is incomplete, by the way), however that study also reports much higher magnitudes.

Response:

We think there are three reasons for this obvious difference:

a. We did not include all regions or all cities across China. Instead, we selected only 231 out of 371 cities in China, which were cities with a total urban land use area larger than a certain size (10 square kilometers).

b. Within these 231 selected cities, we accounted only for urban residents, not the entire population. In total, we accounted for 196 million exposed urban residents out of 1.4 billion people across China. We can do a simple calculation here. Taking China's per capita GDP in 2015 as a reference (\$8066), the total GDP of 196 million people is 1580.9 billion U.S. dollars. So, the

estimated annual economic loss (3 billion) account about 0.19% of total GDP from these exposed population. This result is consisted with the changes in regional GDP across East Asian in 2050 from *Orlov et al. 2020 in Global Env. Change*. For this, we have added more precise calculation process and results in the revised manuscript.

c. Our study does not cover all industries; rather, it covers 16 sectors involving urban economic activity. Other sectors, such as agriculture and forestry, are not considered. Therefore, a significant difference is expected between the calculated sum and the nationwide total.

To clarify this issue in the revised manuscript, first, we have explained that we did not account for all the cities but rather selected cities with a certain area of urban land use (L446-447). Second, we have clarified that this study only accounts for urban residents out of the entire population, as one of our aims is to assess the potential influence of urban adaptation (L407-409). Third, we have revised the presentation of our calculation. For comparison, we calculated the annual total GDPs from all selected urban grids in these 231 cities. The yearly total GDPs were obtained via two steps: 1. Calculate the accounted GDP for each city by multiplying the total urban residents within all selected urban grids by the GDP per capita of the corresponding city. 2. Sum the GDP values of each city to obtain the total accounted GDP (L447-451). Then, we calculated the changes in total GDP declines under RCP6.0 and RCP8.5 compared to the baseline scenario. As shown in Fig. 4, the results are comparable to results across East Asia reported in the cited study, i.e., Orlov et al. 2020 in *Global Env. Change*.

3. Also, the potential of urban adaptation strategies, which are assessed in the study, need to be put into perspective. The study reports that they “could offset more than 0.15 billion” (I assume USD, there was no unit reported). The study again does not discuss the relative magnitude of this major result except for Fig 4b. The adaptation strategies investigated could only offset 5% (or 10% as the 0.15 billion reported in the abstract contradicts Table S2 and Fig 4b) of the economic damage, despite the highly ambitious implementation rates. For instance, the green roof scenario assumes an adaptation rate of 80% across mainland China and the cool-wall scenarios assumes a reflective coating for all wall surfaces in urban areas. The scope and ambition of these scenarios are doubtful, and still produce only a very small impact. Why?

As a reader, one is left wondering why the authors did not focus on this part of the story:

1) Heat stress in Chinese major urban areas causes “just” 3 billion USD in annual economic losses derived from reductions in labor capacity.

2) *Very ambitious adaptation strategies only help to reduce this loss in worker productivity by 5 (or 10%). A possible or even likely reason is that the authors did not reflect about their results this way. If true, this substantially erodes my trust in how rigidly and carefully the analysis has been conducted and interpreted.*

Response:

With respect the first question, we have provided a detailed answer in the last issue.

Regarding the second question, first, we double-checked our assessment results to ensure that there were no calculation errors in the manuscript. As the reviewer points out, there were some unit and data inconsistency problems in the previous version. We have corrected these problems for the different future scenarios. Please see Table S1. Furthermore, we have compared our modeling results with results obtained with the same WRF input and settings, including *Krayenhoff, E. Scott, et al. "Diurnal interaction between urban expansion, climate change and adaptation in US cities." Nature Climate Change 8.12 (2018): 1097-1103* and *Georgescu, Matei, et al. "Urban adaptation can roll back warming of emerging megapolitan regions." Proceedings of the National Academy of Sciences 111.8 (2014): 2909-2914*. We find that the cooling impacts of the adaptation strategies in this study are comparable to those reported in these studies and that the cooling strategies could reduce the average T_{area} by over 0.20 °C while higher average warming of 1–2 °C persists. That is, although these strategies can reduce heat pressure, with a cooling in excess of 1.5 °C at the peak of solar radiation, a gap remains relative to the average warming. To clarify this issue, we have added content to the discussion section; please see L271 to 276.

Furthermore, regarding the cost of application of these adaptation measures, we calculated the additional cost of applying green roofs in the selected urban areas, as the installation of green roofs costs the most among the three selected measures. We calculated the total application cost for all of these cities, which reaches 18.3 billion US dollars. Based on the annual benefits of green roofs, it may take approximately 87 years to recover all the application costs. Many studies have indicated that these building measures can also cool indoor areas, thereby reducing the total energy use and daily costs during the warmest months. Overall, we think the benefit exceeds the application cost in the long run.

4. *There are major results that are unrecognized (or unreflected), which seem to be in conflict with literature and which may point to a flaw in the data analysis and interpretation. Also, the results are taken from just one climate change scenario, without a sensitivity analysis and the adaptation strategy scenarios are highly unrealistic.*

Moreover, crucial aspects are not covered by the analysis. What is the cost-effectiveness of the adaptation measures? The authors seem to implicitly assume no cost, but that is of course not the case. How will technology and the economy react towards increases in heat-stress? What is the sensitivity of results coming from the model's and scenario's parameters and assumptions? I therefore doubt that this is a rigid analysis, which probably requires more than a major revision.

Response

As we noted previously, we have fully considered the comments of the reviewer. The problems can be divided into three areas: 1. the single scenario setting, 2. the significant different total value, and 3. the cost. We have addressed these issues in detail in the last three issues.

5. General remarks: Please use a professional English language editing service. The manuscript is difficult to understand in its content, which to a large part is owed to poor language and syntax. The reader is often left guessing what the authors' intended to say. This makes it quite difficult to understand the details of the analysis. There are also many typos, which can be easily avoided using spell check.

Response:

We have had the revised manuscript edited for English language by American Journal Experts.

6. Remaining major comments: The study's main contribution, beside its empirical findings, is on how in detail the economic structure and population residence is accounted for when analyzing the labor induced heat stress. This is indeed a valuable contribution, but quite technical and not well highlighted. It also does not seem to have a major impact on the results (the % bias it corrects is not explicitly reported, it seems to be less than 10%).

Response:

As we noted above, we have double-checked our assessment results to ensure that there are no calculation errors in the manuscript.

To highlight this part of the results, we have revised the discussion section, as described in our responses to comments. 2 and 3.

7. China may certainly be a very exciting case study, but the reader does not really see why. This could be much better motivated at the beginning (e.g., X% of the world's urban population live in China).

Response:

Some reasons why we selected China as our case study and related information have been added accordingly on L33-36 of the Introduction section.

“This is especially true in China, where more than 2% of the world’s urban population currently live and where more than 70% of the projected population increase will occur in urban areas 7. Therefore, taking cities in China as examples can show the significant impact of continued urbanization on urban warming in the future.”

8. The study does not discuss or consider how cost-effective the adaptation strategies could be. This is a major shortcoming, but could be rectified.

Response:

We have added content to the discussion section, as described in our response to comment 3.

9. I appreciate that the authors try to understand how heat-stress impacts different economic activities at the (sub-)urban level instead of the regional or even national scale. Yet, it remains unclear how the authors estimated the location of the economic activities and how exactly they have treated the seasonality of workloads (e.g., construction workers may not work during the midday heat). Also, such a contribution (while definitely valuable) may not be sufficient for a consideration in a top multi-disciplinary journal. This is also backed by the rather simplistic consideration of other dimensions (economic adjustments are absent, reliance on one SSP, RCP and exposure function, etc).

The economic estimation is very simple and static. There is no endogenous adaptive behavior. Also, no changes in the intensity of the individual economic activities (or even technological improvement). However, the Chinese economy is growing very rapidly and technologies are evolving, so is it reasonable to still assume the same economic structure of today?

Response:

First, we estimated the location of the economic activities. The assessment assumption of our study is based on the latest related studies, such as Zhang Yuqiang and Shindell Drew T. Costs from labor losses due to extreme heat in the USA attributable to climate change. *Climatic Change* 164, 1-18 (2021). In addition, we used a regional climate model to investigate the impact within the urban area.

Second, regarding future economic structures, our study covered 231 cities in total, and the economic structure of each city can be expected to change in a diverse manner in the future. Existing

scenario datasets are unable to support us in completing these detailed assessments, as suggested by the reviewer.

However, we do not assume that economic development across cities will remain unchanged in the future. Our assessment is based mainly on the calculated changes in population exposure. To reflect the changes under future scenarios and the potential impact of future urbanization in China, we used the changes in the population distribution. As we mention in the manuscript, the urbanization rate is projected to grow in the future. In this process, the populations of some large cities will continue to increase while those of some small cities will decrease. Therefore, based on the projected population distribution, we can observe the development of urbanization and economic conditions across different cities, although not the detailed economic structure.

As this point was not discussed in the last version of the manuscript, we have added content addressing this issue in the Discussion section. Additionally, we agree that this is a limitation of our study and have added it to our description of the study limitations.

10. The discussion of how the analysis accounts for changes in the urban population are very difficult to comprehend. I think the authors at least once confuse “urbanization rate” with “share of total population living in urban areas” – see line 236.

Response:

According to the suggestion, we have revised the related content; please see L327-329.

“Regarding urbanization, the percentage of the total population of China that lives in urban areas is 60.9%. The Chinese government expects it to increase to 65.5% by 2030.”

Minor comments:

11. Defining transportation, manufacturing, etc. as medium work-intensity may overestimate heat-stress (also many of these activities can or are done in AC environments - especially until 2050).

Response:

We adopted this type of calculation method after referring to the relevant literature. Currently, there is no research quantifying how the work environments of these sectors will change in the future. Our research hypothesis is more comprehensive than those of other studies that do not consider AC coverage.

We agree this should be a limitation for this study, we have addressed this issue as a limitation in the Discussion section; please see L281 to 283. “We did not consider how AC coverage or

working conditions of these sectors might change in the future, which may have led to some overestimate or underestimate, especially for types of medium-intensity work. Thus, we were unable to precisely estimate the impacts of future economic development other than future population change and changes in working conditions on urban labor loss.”

2. I think the term “labor support” is not helpful for an interdisciplinary audience. I would also recommend to speak of “losses in labor capacity or productivity” than just “labor loss”.

Response:

We have revised the text accordingly (L291-293).

“The expected significant impacts of this increased heat exposure on human health have been studied in detail, while the impact on labor capacity or productivity due to outdoor heat remains unknown. The present study suggests that heat exposure among urban residents will significantly impact labor capacity or productivity across cities in China.”

3. There are many typos, unclear sentences, strange formatting, etc. - this gives the reader a sloppy impression.

Response:

We have had the revised manuscript edited for English language by American Journal Experts.

Reviewer #3:

This study assesses the heat-induced impacts on labour productivity and associated economic costs under different adaptation measures at a sub-urban scale. The results from this impact assessment provide some novel and useful insights. Especially, I find interesting the combination of a regional climate model with an urban canopy model to investigate the effectiveness of different adaptation options in urban areas. I have a few questions and improvement suggestions:

1. In Lines 205-206, the authors write: "First, we established only one future climate-society scenario, the modelling results would be slightly different if we compared the results from scenarios under different assumptions". I am not sure that the results would be 'slightly' different. The uncertainties associated with climate modelling (i.e., climate sensitivity) are large, so are socio-economic uncertainties. I guess the authors did not have enough capacity to conduct a dynamic downscaling for multiple RCPs and climate models. However, I wonder if it makes sense to consider, two or three SSPs to investigate the relevance of socio-economic uncertainties.

Response:

In the revised manuscript, the comments of the reviewer have been fully considered. We have added another climate change scenario, RCP 8.5, and compared the results with our previous results, and our results and discussion are now more reliable.

We also compared the labor and following economic losses between the two climate change scenarios. The results show that the economic losses under the two different scenarios are comparable but that the spatial pattern differs between these two scenarios due to the different warming patterns.

2. I am a bit surprised that the economic cost from construction is so large compared to other sectors. I understand that construction is more labour-intensive and requires working outdoors, but I guess much more people are working in manufacturing and services. So, I would expect that the total cost of heat-induced impacts on labour productivity in manufacturing and services would be larger than in construction. On that regard, I think it would be useful to have a table or figure showing the sectoral wage rates and the number of people employed in each sector and region. Also, please report the recent penetration of air conditioners by regions.

Response:

We think the reasons of the biggest cost from construction is not only because of the high labor density, but also the completely exposure to the high temperature conditions without any coverage of air conditioning. Based on the assumption:

$$CPop_i^j = Pop_i^j \times (1 - C^j)$$

$CPop_i^j$ represents the total number of residents of each urban grid from city j employed by sector i who are not protected by AC. C^j represents the penetration rate of AC in city j for all sectors except the construction sector, as it mainly entails outdoor work. So, the total exposed population from construction would be higher than other industries, which leads to a larger total cost of heat-induced impacts on construction. It is consistent with the related study: Orlov Anton, Sillmann Jana, Aunan Kristin, Kjellstrom Tord, Aaheim Asbjørn. Economic costs of heat-induced reductions in worker productivity due to global warming. *Global Environmental Change* 63, 102087 (2020).

Furthermore, we have added the Table S3, which details the sector wage rates, the ratio of people employed in each sector and the recent penetration of air conditioners by region.

3. I would expect that the economic structure will change over time (i.e., more income will be generated from the service sector compared to manufacturing and construction). However, it seems that a structural change in the regional economies is not implemented in this study. In this case, this should be mentioned as a limitation.

Response:

Our assessment is based mainly on the calculated changes in population exposure. To reflect the changes under future scenarios and the potential impact of future urbanization in China, we used the changes in the population distribution. As we mention in the manuscript, the urbanization rate is projected to grow in the future. In this process, the populations of some large cities will continue to increase while those of some small cities will decrease. Therefore, based on the projected population distribution, we can observe the development of urbanization and economic conditions across different cities, although not the detailed economic structure.

We agree that this is a limitation of our study, and we have discussed it as such in the revised manuscript (L277-281).

4. Also, there are other adaptation measures, which could be relevant to urban areas. For example, mechanization in construction, shifting working hours, acclimatisation etc. The authors should mention this in the discussion.

Response:

Discussion of these adaptation measures has been added accordingly. Please see L274 to 276.

5. Regarding the presentation of economic impacts, I find it useful to show not only the total cost but also relative impacts, such as the cost per worker (e.g., relative changes in income per worker or USD lost per worker due to heat stress). One could also show the cost per worker in comparison to the income per worker in different regions to reveal potential impacts on income inequality and distribution across regions.

Response:

We agree with the opinions of experts, but in consideration of suggestions by Reviewer 2 and to facilitate comparisons with related studies, we consider it preferable to calculate the changes in total GDP loss under RCP6.0 and RCP8.5 with SSP2 compared to the baseline scenario, as shown in Fig. 4.

6. Fig. 4 shows that in some regions, the annual economic loss could be larger than 200%, while the labour productivity loss does not exceed 8%. The authors should better explain the calculation of economic losses. The calculated economic costs seem to be implausible large.

Response:

As we described in our response to the previous comment, to clarify the results according to the reviewer's suggestion, we calculated the changes in total GDP loss under RCP6.0 and RCP8.5 with SSP2 compared to the baseline scenario, as shown in the revised Fig. 4. In addition, we compared these results to those of related research. We found that the total GDP account is consistent with that reported in a related study.

7. As far as I know, the Bernard & Pourmoghani's method requires an iterative solution, but the method for WBGT calculation, which is described in Supplementary Information, does not seem like an iterative method. Please clarify it. Do you use an original Bernard & Pourmoghani's method or its simplification? Also, please add the formula for calculation of the dewpoint temperature.

Response:

We calculated the WGBT according to the related study: Andrews, Oliver, et al. "Implications for workability and survivability in populations exposed to extreme heat under climate change: a modelling study." *The Lancet Planetary Health* 2.12 (2018): e540-e547.

We have described the calculation method of dewpoint temperature in the revised Supplementary materials

8. I would suggest that the manuscript should undergo extensive English revisions because there are many typos and errors.

Response:

We have had the revised manuscript edited for English language by American Journal Experts.

REVIEWER COMMENTS

Reviewer #1 (Remarks to the Author):

General Comments:

A good paper about a much-needed study topic - The author deserves praise for focusing on a crucial topic that receives little attention. The necessary modifications have now been implemented. The updated manuscript is written with clarity in several areas, making it understandable to the readers. Except for one, the authors have done a fantastic job of studying and responding to all of the suggested comments.

Specific comments:

1. Comment no 19 has not been addressed in the manuscript - Add about the ethical approval, if any as per your country guidelines.

Reviewer #2 (Remarks to the Author):

I would like to thank the authors for their efforts in addressing my concerns and questions. Most of my points were clarified.

I appreciate the efforts on estimating the cost of the adaptation measures. Yet, here, I do not agree with the authors' assessments (see point 3 below).

I have the following remaining comments:

1. I think one source of confusion that makes it difficult to put the findings into relation (from an economic point of view) is the missing clarity on what population (in million people and in % of total urban population in China) is covered by this study. The study mentions 231 cities, the rebuttal letter also mentions that 196 million people were covered. I think it would be better if the manuscript itself is more explicit here (231 cities of a total of X cities, 196 million urban residents of a total of X million urban residents).

2. I now better understand that your study also seems to study the effects of "urban crowding" (see L35), which you show by calculating Tarea and Tpop, where the latter is the regional population-weighted WBGT.

Could you please clarify if you also model a relationship between density of urban areas and the climate-change induced change in WBGT? Or do you rather observe that the absolute impact of heat stress (WBGT) is increasing with higher shares of population? I think the manuscript implies the former, but from my reading the actual estimation does the latter. If the former is the case, then I miss a description of how exactly the change in WBGT is dependent on changes in urban crowding.

3. I think the ms benefits greatly from an improved presentation of the economic costs and benefits. However, the added paragraph on the costs of adaptation measures gives the impression that the adaptation measures are cost-effective. First, it is difficult to trace the detailed calculations, which should be provided in the appendix. Second, I am afraid that the calculation is not based on sound economic principles, which accounts for the cost of capital and time value of money (i.e., discounting future benefits by a (social) discount rate) as it is typically done for such capital-intensive interventions. The authors' present a payback period of 87 years for the adaptation measure "green roofs" (but not for the remaining measures) and it seems that no discount rate was applied. This very long payback period suggests that the adaptation measures do not seem to be cost-effective. Typically, investment projects with a discount rate of about 5% and constant annual benefits have a payback period of about 6-8 years. A payback period of 87 years would imply a discount rate of <0.1%. This does not demonstrate economic viability, as no public or private investor would (rationally) choose to invest funds in such a project. Furthermore, the analysis did not consider the

operation and maintenance costs of green roofs, which are likely to be non-trivial (but the authors also rightly emphasize that there could be energy savings due to green roofs). The authors have to be much more careful in their assessment in this regard.

4. As also mentioned in my comments in the first round, I doubt that an 80% adoption rate for the adaptation measures is far from realistic. I think the paper should better reflect on this assumption.

Minor comments

L 33: "this is especially true in China, where more than 2% of the world's urban population currently live .." I doubt that it is just 2%. About 60% of China's population resides in urban areas or about 0.8 billion people. There are 7.8 billion people in the world, so that share should be >10%.

L48: How exactly is the urban economy affected?

L68: A closing ")" is missing.

L131: "Labor losses" is odd. It should be "changes in labor productivity" or "..capacity"

L169: There seems to be a typo with transportation

L403: "Orlov used this function to calculate the market costs of future global warming-induced reductions in labor loss worldwide." I think it should rather read "reductions in labor productivity"

Reviewer #3 (Remarks to the Author):

i find the authors revision sufficient and recommend the acceptance of the manuscript after correcting Fig. 1. Please change "RCP4.5" to "RCP6.0" on the left side of the figure.

Reviewer #1

A good paper about a much-needed study topic - The author deserves praise for focusing on a crucial topic that receives little attention. The necessary modifications have now been implemented. The updated manuscript is written with clarity in several areas, making it understandable to the readers. Except for one, the authors have done a fantastic job of studying and responding to all of the suggested comments.

Specific comments:

1. Comment no 19 has not been addressed in the manuscript - Add about the ethical approval, if any as per your country guidelines.

Response:

We apologize that the previous responses did not clearly explain this issue. In our study, no research content occurs involving human subjects. Therefore, there was no need to obtain ethical approval.

Reviewer #2 (Remarks to the Author):

I would like to thank the authors for their efforts in addressing my concerns and questions. Most of my points were clarified. I appreciate the efforts on estimating the cost of the adaptation measures. Yet, here, I do not agree with the authors' assessments (see point 3 below). I have the following remaining comments:

1. I think one source of confusion that makes it difficult to put the findings into relation (from an economic point of view) is the missing clarity on what population (in million people and in % of total urban population in China) is covered by this study. The study mentions 231 cities, the rebuttal letter also mentions that 196 million people were covered. I think it would be better if the manuscript itself is more explicit here (231 cities of a total of X cities, 196 million urban residents of a total of X million urban residents).

Response:

This missing content has been added accordingly (lines 459–460).

“Overall, this study involved 231 cities out of a total of 333 Chinese cities and 196 million urban residents out of a total of 643 million urban residents.”

2. I now better understand that your study also seems to study the effects of “urban crowding” (see L35), which you show by calculating T_{area} and T_{pop} , where the latter is the regional population-weighted WBGT. Could you please clarify if you also model a relationship between density of urban areas and the climate-change induced change in WBGT? Or do you rather observe that the absolute impact of heat stress (WBGT) is increasing with higher shares of population? I think the manuscript implies the former, but from my reading the actual estimation does the latter. If the former is the case, then I miss a description of how exactly the change in WBGT is dependent on changes in urban crowding.

Response:

We do consider the impact of future urbanization on WBGT change when we modeled urban climate conditions under specific future climate change scenarios.

Specifically, to reflect the attributes of future urbanization, we updated the land-use settings of the WRF model based on expanded urban land-use data under the SSP2 scenario so as to combine future urbanization and climate change impacts in the simulation of WBGT changes under climate change scenarios.

To better explain this issue as suggested, we have added an additional explanation in the Methods section; please refer to lines 354–357.

“We updated the land-use settings of the WRF model based on these urban land-use and population data (please refer to the Appendix for details) and combined it with climate change scenarios in the simulation of WBGT changes.”

3. I think the MS benefits greatly from an improved presentation of the economic costs and benefits. However, the added paragraph on the costs of adaptation measures gives the impression that the adaptation measures are cost-effective. First, it is difficult to trace the detailed calculations, which should be provided in the appendix. Second, I am afraid that the calculation is not based on sound economic principles, which accounts for the cost of capital and time value of money (i.e., discounting future benefits by a (social) discount rate) as it is typically done for such capital-intensive interventions. The authors' present a payback period of 87 years for the adaptation measure "green roofs" (but not for the remaining measures) and it seems that no discount rate was applied. This very long payback period suggests that the adaptation measures do not seem to be cost-effective. Typically, investment projects with a discount rate of about 5% and constant annual benefits have a payback period of about 6-8 years. A payback period of 87 years would imply a discount rate of <0.1%. This does not demonstrate economic viability, as no public or private investor would (rationally) choose to invest funds in such a project. Furthermore, the analysis did not consider the operation and maintenance costs of green roofs, which are likely to be non-trivial (but the authors also rightly emphasize that there could be energy savings due to green roofs). The authors have to be much more careful in their assessment in this regard.

Response:

First, we have added detailed calculations in the Appendix as suggested; please refer to the Supplementary Methods.

Second, as pointed out by the reviewer, we did not consider the cost of capital and time value of money in the calculation of the cost of adopting green roof systems. Because our study also did not consider the capital cost and time value of money in the calculation of the recovered economic losses due to the implementation of green roofs. As we described in the limitations of the study, we could not quantify the projected changes in work income per hour in the different cities with the time value of money. So, these effects are not considered in this part in order to keep all research results consistent.

In addition, based on relevant research (What are the root causes hindering the implementation of green roofs in urban China? *Science of the Total Environment* 654, 742-750 (2019)), we replaced the original green-roof building costs with suggested construction standard costs obtained from the Beijing government and found that 20 years would be required to recover all construction costs.

But we agree with the reviewer that we simplified the quantification calculation in this part, without considering daily maintenance costs, cost of capital and time value of money in the recovery process, which should be evaluated with a detailed economic model. As such, we have added additional explanations (lines 264 to 267). Based on the current assessment results and potential reduction in the daily energy use of green roofs, we believe that the benefits of applying green roofs exceed the application costs in the long run.

4. As also mentioned in my comments in the first round, I doubt that an 80% adoption rate for the adaptation measures is far from realistic. I think the paper should better reflect on this assumption.

Response:

First, based on existing policies and the status quo, such as the Shanghai Municipal Greening Regulations requiring all newly built public buildings to adopt green roofs,

the requirements that the total coverage of green roofs must not be lower than 50%, and the addition of more than 420000 m² of green roofs since 2010 in Hangzhou, we expect that cities in China will exhibit high green roof coverage levels in the future. Second, although there exist certain gaps between the current green roof coverage state and our modeling settings, our goal was to demonstrate the potential impact of a feasible near-future scenario involving the upper bound of climate effects based on relevant studies (Krayenhoff E Scott, Moustou Mohamed, Broadbent Ashley M, Gupta Vishesh, Georgescu Matei. Diurnal interaction between urban expansion, climate change and adaptation in US cities. *Nature Climate Change* 8, 1097-1103 (2018).). Therefore, under our green roof scenario, we set the adoption rate to 80%.

Minor comments

1. L 33: "this is especially true in China, where more than 2% of the world's urban population currently live .:" I doubt that it is just 2%. About 60% of China's population resides in urban areas or about 0.8 billion people. There are 7.8 billion people in the world, so that share should be >10%.

Response:

We have corrected this part of the statement with data retrieved from the World Bank (https://data.worldbank.org/indicator/SP.URB.TOTL.IN.ZS?name_desc=false); please refer lines 33 to 34.

"This is especially true in China, where more than 19.74% of the world's urban population currently lives, and more than 70% of the projected population increase will occur in urban areas ⁷."

2.L48: How exactly is the urban economy affected?

Response:

Due to their varied total populations, age patterns, and economic structures, the adaptive capacities of cities to climate change differ and have not been fully considered in current studies. Please refer lines 48 to 50.

3.L68: A closing “)” is missing.

Response:

The missing “)” has been added in line 68.

4. L131: “Labor losses” is odd. It should be “changes in labor productivity” or “..capacity”

Response:

We have revised this statement accordingly; please refer to line 135.

5.L169: There seems to be a typo with transportation

Response:

We have revised the text; please refer to line 175.

6. L403: “Orlov used this function to calculate the market costs of future global warming-induced reductions in labor loss worldwide.” I think it should rather read “reductions in labor productivity”

Response:

We have revised that statement accordingly; please refer to lines 414 to 416.

Reviewer #3 (Remarks to the Author):

I find the authors revision sufficient and recommend the acceptance of the manuscript after correcting Fig. 1. Please change "RCP4.5" to "RCP6.0" on the left side of the figure.

Response:

We have revised the text; please refer to line 94.

REVIEWER COMMENTS

Reviewer #2 (Remarks to the Author):

I thank the authors for their responses and clarifications. I have no further questions nor queries.

Reviewer #3 (Remarks to the Author):

i find the authors revision sufficient and recommend the acceptance of the manuscript after correcting Fig. 1. Please change "RCP4.5" to "RCP6.0" on the left side of the figure.

Reviewer #2

I thank the authors for their responses and clarifications. I have no further questions nor queries.

Response:

We thank the reviewer's valuable suggestions.